# Multivariate Prognostic Model for Predicting the Outcome of Critically Ill Patients Using the Aromatic Metabolites Detected by Gas Chromatography-Mass Spectrometry

**DOI:** 10.3390/molecules27154784

**Published:** 2022-07-26

**Authors:** Alisa K. Pautova, Andrey S. Samokhin, Natalia V. Beloborodova, Alexander I. Revelsky

**Affiliations:** 1Negovsky Research Institute of General Reanimatology, Federal Research and Clinical Center of Intensive Care Medicine and Rehabilitology, 25-2 Petrovka Str., 107031 Moscow, Russia; nvbeloborodova@yandex.ru; 2Chemistry Department, Lomonosov Moscow State University, GSP-1, Leninskie Gory 1-3, 119991 Moscow, Russia; andrey.s.samokhin@gmail.com (A.S.S.); sorbent@yandex.ru (A.I.R.)

**Keywords:** sepsis, SOFA, APACHE II, 4-hydroxyphenyllactic acid, SIMCA, data analysis

## Abstract

A number of aromatic metabolites of tyrosine and phenylalanine have been investigated as new perspective markers of infectious complications in the critically ill patients of intensive care units (ICUs). The goal of our research was to build a multivariate model for predicting the outcome of critically ill patients regardless of the main pathology on the day of admission to the ICU. Eight aromatic metabolites were detected in serum using gas chromatography-mass spectrometry. The samples were obtained from the critically ill patients (*n* = 79), including survivors (*n =* 44) and non-survivors (*n* = 35), and healthy volunteers (*n* = 52). The concentrations of aromatic metabolites were statistically different in the critically ill patients and healthy volunteers. A univariate model for predicting the outcome of the critically ill patients was based on 3-(4-hydroxyphenyl)lactic acid (*p*-HPhLA). Two multivariate classification models were built based on aromatic metabolites using SIMCA method. The predictive models were compared with the clinical APACHE II scale using ROC analysis. For all of the predictive models the areas under the ROC curve were close to one. The aromatic metabolites (one or a number of them) can be used in clinical practice for the prognosis of the outcome of critically ill patients on the day of admission to the ICU.

## 1. Introduction

Metabolomic approaches are actively used in the search, evaluation, and clinical verification of the biomarkers of the critical states. Infectious complications, including sepsis, remain a serious life-threatening conditions in critically ill patients with different pathologies. The science community has recognized the need to search for specific markers for the verification of the infectious complications in critically ill patients [1]. The absence of a positive blood culture is a common phenomenon and the decision to prescribe antibiotics is often based on the different international clinical scales, such as Sequential Organ Failure Assessment (SOFA) [2] and Acute Physiology and Chronic Health Evaluation II (APACHE II) [3], and recommendations, such as the Third International Consensus on the Definition of Sepsis and Septic Shock (SEPSIS-3) [4].

Lactate, procalcitonin, C-reactive protein, interleukin-6, and some other compounds are known to be non-specific infectious biomarkers and are widely used in clinical practice. However, new trends in the sepsis treatment include personalized approaches based on the specific markers with high specificity, sensitivity, and response to the treatment [5,6]. A number of aromatic metabolites of tyrosine and phenylalanine, including those of microbial origin, have been investigated as perspective sepsis-specific markers during the last two decades (Table 1) [7,8,9]. The main hypothesis was formulated that the accumulation of aromatic metabolites in the blood of patients with sepsis was the result of the microbial degradation of phenylalanine and tyrosine [10]. Experimental studies demonstrated their pathophysiological role in sepsis development [11]. Clinical studies revealed their diagnostic significance in sepsis and septic shock caused by various pathologies [12,13], such as community-acquired pneumonia [14], acute surgical diseases of the abdominal organs [15], acute or chronic (end stage) renal failure [16], postoperative cardiosurgical complications [17], and post-neurosurgical meningitis [18]. There are positive correlations between the sepsis-associated aromatic metabolites phenyllactic, *p*-hydroxyphenylacetic, and *p*-hydroxyphenyllactic (*p*-HPhLA) acids and the clinical scales (SOFA and APACHE II), serum lactate, and procalcitonin [11,14,15]. In particularly, the dynamics of the total concentration of sepsis-associated aromatic metabolites in serum appeared to be a more accurate biochemical criterion of the therapy’s efficiency than lactate dynamics in patients with acute surgical abdominal diseases [15]. In total, the results of the clinical studies indicated that the aromatic compounds could be diagnostic markers, regardless of the main pathology of the critically ill patients [12,19,20].

Most of the clinical studies listed above [13,14,15,16] were carried out using gas chromatography with flame ionization detection. The detection limits were restricted by the low selectivity of this method and the significant matrix effect which caused the increase of the baseline level. The use of a mass selective detector improved the analytical performance and led to the detection of the extended list of compounds, including benzoic, phenylpropionic, *p*-hydroxybenzoic, homovanillic, and *p*-hydroxyphenylpropionic acids [21].

For the practical use of the aromatic metabolites as a diagnostic tool, it is important to find out if one or a number of them demonstrates any predictive value for the outcome of critically ill patients. Previous clinical studies have described the univariate predictive models based on concentrations of one or a sum of several metabolites (phenyllactic, *p*-hydroxyphenylacetic acids, and *p*-HPhLA) [13,15,18]. In addition, it is important to reveal if these compounds could be used as predictive markers on the day of admission to the intensive care unit (ICU), by analogy with the APACHE II scale, thus, indicating patients with more severe disease and a higher risk of death. The main goal of our research was to build a multivariate model for predicting the outcome of critically ill patients regardless of the main pathology on the day of admission to the ICU, using serum levels of eight aromatic metabolites detected by gas chromatography-mass spectrometry (GC-MS).

## 2. Results

### 2.1. Characteristics of the Critically Ill Patients

The critically ill patients were characterized by the outcome (survivors/non-survivors); the oxygenation index (an indicator of the impaired respiratory function); the prescription of vasopressors (an indicator of the heart dysfunction); and the Glasgow Coma Scale value (an indicator of the central nervous system dysfunction), SOFA (a scale of the patient’s status during the stay in the ICU), and APACHE II (a severity-of-disease classification system within 24 h of admission to the ICU) scores. The serum samples were characterized by the day of sampling; platelet cell concentration (an indicator of the dysfunction of the rheological properties of blood); total bilirubin (an indicator of the impaired liver function); creatinine (an indicator of the impaired kidney function). All of the patients were divided into two groups as survivors (*n* = 44) and non-survivors (*n* = 35). The description of the two groups of critically ill patients is summarized in Table 2.

Eight aromatic metabolites were measured in all of the serum samples (*n* = 196) and their occurrence was different. Benzoic acid and *p*-HPhLA were detected in all of the samples: phenyllactic; *p*-hydroxyphenylacetic; phenylpropionic; homovanillic; *p*-hydroxybenzoic; and *p*-hydroxyphenylpropionic acids were detected in 81, 78, 48, 31, 17, and 9% of the serum samples, accordingly. Statistically significant differences (*p* < 0.05) were found between all of the groups (healthy volunteers, survivors, and non-survivors) for all of the metabolites (Table 3).

### 2.2. Multivariate Classification Model Based on Eight Aromatic Metabolites

The concentrations of some of the metabolites (e.g., homovanillic, *p*-hydroxyphenylacetic acids, or *p*-HPhLA) varied significantly among the non-survivors (Table 3). On the contrary, the samples obtained from the survivors were less spread out. It is additionally illustrated in Appendix A, where two metabolites (i.e., homovanillic acid and *p*-HPhLA) are considered. The samples obtained from the survivors constitute a compact group near the origin. On the other hand, the samples from the non-survivors are significantly scattered (Appendix A). If both of the classes formed compact groups in the multidimensional space, discriminant techniques (such as partial least squares discriminant analysis) could be applied to model both of the classes. However, in our research, the members of the “Non-survivors” class are different from each other and they are spread in multidimensional space. For this reason, it was decided to build a one-class classification model. Moreover, it gives an additional benefit when the number of samples is limited: the samples obtained from the non-survivors are not used during the construction of the model and, as a result, they should not be split into calibration and test sets.

As can be seen from Table 3 and Appendix A, the concentration of phenylpropionic acid is higher in the healthy volunteers than in the survivors. As a result, several healthy volunteers with a high level of phenylpropionic acid would be classified as “Non-survivors”, if the classification model was built using only the survivors. Merging the healthy volunteers and the survivors into one class solves this problem. Additionally, it significantly increases the number of samples in the target class.

If the concentration of at least one of the metabolites was atypically high, a sample was found to be an outlier. Examining the concentration distributions and applying multivariate approaches [22] showed that 15 samples in the group of survivors were outliers. Seven of them were replaced with samples taken a few days after admission from the same patients. However, in eight cases it was not possible, because either only one sample was taken from a particular patient or all of the samples were found to be outliers (Appendix A). These samples were not used during the calibration and validation steps and formed an additional group which was called “Survivors (outliers)”.

For some members of the “Non-survivors” class, the concentration of one or more of the analytes exceeded the maximum concentration of the calibration curve. For this reason, relative areas were used to build multivariate classification models, using soft independent modeling by class analogy (SIMCA).

The distance plots are typically used to visualize the results of SIMCA classification (Figure 1a,b). To build such a plot, the orthogonal and score distances are calculated for each sample and all of the samples are projected on the respective plane. In this work, the distances were log-transformed to make it possible to display all of the samples on the same plot. The acceptance area (also known as cut-off value or decision rule) was calculated, using a data-driven approach employing the assumption that the distances followed the scaled chi-squared distribution [23]. The decision rule is represented by a black solid line on the distance plot.

It should be emphasized that the approach to choosing the cut-off value differs from the two-class classification methods. When both classes are modeled, sensitivity and specificity are simultaneously examined. There are several criteria for choosing the optimal cut-off value [24]. In contrast, we used a one-class classification method in combination with the data-driven approach. It means that the cut-off value is calculated using only one class. The cut-off value can be adjusted by setting an appropriate significance level (α).

In this work, we also implemented another visualization approach. It is based on calculating the full distance [22]. The full distance represents both the value of the residuals (i.e., the orthogonal distance) and the distance to the center of the model (i.e., the score distance). From a geometrical point of view, it is equivalent to projecting all of the points onto a straight line. We believe that such plots are clearer and more compact when several groups are considered simultaneously (Figure 1c). In the case of such one-dimensional plots, the decision rule is represented by a vertical line.

It can be seen from Figure 1c that one sample from the calibration set was misclassified, i.e., it was to the right of the decision rule line. The decision rule was calculated from the data for the significance level α = 0.01, using a data driven approach. Thus, some samples from the target class can be misclassified. As long as the number of such points is comparable to the given significance level, the model works adequately.

The false-negative rates for both the calibration and the Procrustes cross-validation were equal to 0.017, which are close to the selected significance level α = 0.01. It should be emphasized that the test set was used only for the evaluation of the model efficiency. All of the samples from the test set were classified correctly.

All of the samples which appeared to be outliers and formed the “Survivors (outliers)” group were to the right of the decision rule line. The rest of the samples from the survivors, which were not used either in the calibration or the test group, formed the “Survivors (others) group”. As can be seen from Figure 1c, such samples are located on both sides of the vertical line. This behavior can be explained by changing the profile of the metabolites in the patients during their stay in the ICU.

Only one sample from the “Non-survivors” group was misclassified. It happened because the concentrations of all of the metabolites were in the range typical for the calibration and test sets. Unfortunately, only one sample was taken from this patient, and it is not possible to investigate changes in metabolomic profile during treatment in the ICU.

### 2.3. Univariate Classification Models

It was shown in the previous studies [13,15] that *p*-HPhLA can be used to predict the outcome of critically ill patients. It was interesting to compare the univariate models based on the concentration of the only *p*-HPhLA, the clinical APACHE II scale, and the multivariate model (based on the concentrations of all eight of the metabolites). The results obtained for the univariate models are summarized in Figure 2. The “Calibration” and “Test” groups were merged into one group, “Healthy volunteers and survivors”, for the univariate model based on the concentration of the only *p*-HPhLA, because the univariate model does not require the optimization of any free parameter (the decision rule was estimated using a data-driven approach, without taking into account the “Non-survivors” group for α = 0.01).

The univariate model based on the concentration of the *p*-HPhLA with the decision rule of 4.1 µmol/L demonstrates that all of the samples from the modeling class were classified correctly, but three samples from the “Non-survivors” and three samples from the “Non-survivors (others)” groups were misclassified. For the univariate model based on the clinical APACHE II score with the decision rule of 19 points (negative prognosis begins from 20 points), all of the samples from the modeling class were classified correctly, but 13 and 5 samples from the “Non-survivors” and “Non-survivors (others)” groups were misclassified, respectively.

### 2.4. Multivariate Classification Model Based on Seven Aromatic Metabolites

A comparison of the results obtained using the univariate and multivariate models raises one more question: whether the other metabolites (i.e., except *p*-HPhLA) have any predictive value. To shed light on this issue, we excluded the *p*-HPhLA from consideration and rebuilt the SIMCA model. The results are summarized in Figure 3. Excluding *p*-HPhLA had only a small effect on the results. The modeled class containing both the healthy volunteers and survivors, and the “Non-survivors” group became less separated. As a result, two of the samples from the “Calibration” group were misclassified and three samples from “Non-survivors” and “Non-survivors (others)” groups were inside the acceptance area.

### 2.5. ROC Analysis

Receiver operating characteristic (ROC) analysis was applied to estimate the overall performance of the models (Figure 4). The “Calibration” and “Test” groups were used to calculate the sensitivity and the “Non-survivors” group was used to calculate the specificity. The area under the ROC curve is typically used to summarize the overall diagnostic accuracy of the models [25]. For all of the models, the areas under the ROC curve were close to one (Figure 4). In the case of the multivariate models, the area under the curve was smaller in comparison with the univariate model, based on the concentration of the only *p*-HPhLA (AUC = 0.990). However, the areas under the ROC curve for all of the models based on the aromatic metabolites (AUC ≥ 0.978) were higher than for the clinical APACHE II scale (AUC = 0.966).

## 3. Discussion

In this research, we built several prognostic models which could successfully predict the outcome of critically ill patients, regardless of the main pathology on the day of admission to the ICU. The target compounds for the prognostic models were phenyl-containing acids. Some of them (sepsis-associated phenyllactic, *p*-hydroxyphenylacetic acids, and *p*-HPhLA) were considered previously in the univariate models of the outcome for the patients with similar pathologies, i.e., in homogeneous cohorts. Summarizing the results of the previous clinical studies, we formulated the following hypothesis: aromatic metabolites could be prognostic markers of the patient’s outcome regardless of the main pathology.

The most important compound, *p*-HPhLA, is a metabolite of tyrosine. It was used to build the univariate model. Other metabolites of tyrosine, such as *p*-hydroxyphenylacetic, homovanillic, *p*-hydroxyphenylpropionic, and *p*-hydroxybenzoic acids, and metabolites of phenylalanine, such as phenylpropionic, hydroxyphenyllactic, and benzoic acids were considered simultaneously to construct two multivariate models. One of them included all eight of the mentioned metabolites; the other one did not include *p*-HPhLA. All of the models demonstrated satisfactory performance according to the area under the ROC-curve (AUC ≥ 0.978).

The main difference between the multivariate and univariate models is the prediction obtained for the samples with atypical concentrations of one or several metabolites. Such samples formed the “Survivors (outliers)” group for the SIMCA models (Figure 1 and Figure 3). However, most of these samples have relatively low concentrations of *p*-HPhLA which is typical for the “Survivors” group (Figure 2a). It is an open question as to why there is such a difference. On the one hand, the multivariate model may be more sensitive. On the other hand, changes in the concentration of the metabolites other than *p*-HPhLA can indicate other conditions which were not connected with the outcome. Thus, the univariate model based on *p*-HPhLA is able to classify correctly more survivors (less false-positive samples) than the multivariate models.

At the same time, the univariate model gave an incorrect prediction for the higher number of non-survivors. One sample from the “Non-survivors” group in the multivariate model based on eight metabolites, and three samples from the “Non-survivors” and “Non-survivors (others)” groups in the multivariate model based on seven metabolites (Figure 3) were misclassified, in contrast to six samples from the “Non-survivors” and “Non-survivors (others)” groups in the univariate model based on *p*-HPhLA. Considering the possible application of the prognostic models in clinical practice, the use of the univariate model will result in a smaller number of patients in the “risk” group and more chance to “miss” the patients with a poor outcome. Hence, the multivariate models will result in more patients in the “risk” group and a lower mortality rate in the ICU. The latter, in turn, will increase the cost of the patient’s treatment. Thus, the “golden mean” is a subject for subsequent discussion.

Despite the more false-negative samples for the univariate model, the areas under the ROC curve for this model were the highest in contrast to the multivariate models. The main reason was a sample in the “Non-survivors” group for the multivariate models having a relatively small full distance. If this sample were located closer to the vertical line in Figure 1 and Figure 3, the areas under the curve would be higher.

It is more significant that the areas under the ROC curve for all of the models based on the aromatic metabolites (AUC ≥ 0.978) were higher than for the clinical APACHE II scale (AUC = 0.966). The decision rule for the univariate model based on the clinical APACHE II score was 19 points and negative prognosis began from 20 points. This scale has been used since the 1980s to assess the ICU mortality and has been tested many times around the world [26]. There are special calculators which are used to obtain the APACHE II score points and to find the corresponding mortality rate [27]. At 15–19 points, the approximated in-hospital mortality rates are 25% for nonoperative and 12% for postoperative patients; at 20–24 points, the approximated in-hospital mortality rates are 40% for nonoperative and 30% for postoperative patients. Thus, a patient with a score of 20 is already in a “risk” group and the use of any predictive model based on aromatic metabolites would help in further patient stratification.

Moreover, our results suggest that severe metabolic disturbances in critically ill patients on the day of admission to the ICU are as important as other clinical parameters, such as oxygenation and blood pressure, and should also be treated. In the case of aromatic metabolites, this treatment should include properly selected antibiotic therapy and monitoring of its effectiveness.

## 4. Materials and Methods

### 4.1. Reagents and Standards

The benzoic acid (≥99.5%), 2,3,4,5,6-D_5_-benzoic acid (internal standard, ≥99 atom % D, ≥99%), 3-phenylpropionic acid (phenylpropionic, ≥99%), 3-phenyllactic acid (phenyllactic, ≥98%), 4-hydroxybenzoic acid (*p*-hydroxybenzoic, ≥99%), 2-(4-hydroxyphenyl)acetic acid (*p*-hydroxyphenylacetic, ≥98%), 4-(3-hydroxyphenyl)propionic acid (*p*-hydroxyphenylpropionic, ≥98%), homovanillic acid (≥97%), 4-(3-hydroxyphenyl)lactic acid (*p*-HPhLA, ≥97%), *N*,*O*-bis(trimethylsilyl)trifluoroacetamide (99%, contains 1% trimethylchlorosilane), hexane (≥97.0%) were obtained from Merck (Darmstadt, Germany); sulfuric acid, diethyl ether, sodium chloride were Laboratory Reagent grade and obtained from Khimreactiv (Staryy Oskol, Russia).

### 4.2. Sample Preparation and GC-MS Analysis

The sample preparation and GC-MS analysis conditions of the serum samples were described in detail in the previous study [21]. Briefly, the sample preparation of the serum samples included dilution with an aqueous solution of internal standard (4 mg/L) and distilled water, protein precipitation using sulfuric acid and solid sodium chloride, double liquid–liquid extraction with diethyl ether, evaporation, silylation, and dilution with hexane before GC-MS analysis.

The GC-MS analysis was performed on a Trace GC 1310 gas chromatograph with helium as the carrier gas (1.5 mL/min) using the capillary column TR-5ms (95% poly(dimethylsiloxane) + 5% phenyl polysilphenylene-siloxane phase, 30 m × 0.25 mm, df = 0.25 µm) equipped with an ISQ LT mass spectrometer and AI 1310 autosampler, obtained from Thermo Scientific (Thermo Scientific, Thermo Electron Corporation, Waltham, MA, USA). The conditions of the GC analysis were the following: split 1:10; inlet temperature 260 °C; injection volume 2 µL; initial temperature 80 °C with ramp 10 °C/min to 250 °C and hold time 4 min; time of GC analysis 27 min. The conditions of the MS analysis were the following: electron-impact mode 70 eV; *m*/*z* range: 50–480; scan rate 3 scans/s; MS source 200 °C; GC-MS interface 250 °C; software Xcalibur 2.2.

The trimethylsilyl derivatives of the analytes were identified using the retention times and characteristic *m*/*z* of the derivatized standards. The quantitative analysis was carried out using calibration curves and internal standard (2,3,4,5,6-D_5_-benzoic acid). The linear analytical range for the analytes was 0.5–36 µmol/L. The relative standard deviation (RSD) of LOQ value (0.5 µmol/L in average) was 10–12% for benzoic, phenylpropionic, phenyllactic acids, and *p*-HPhLA; and 18–30% for *p*-hydroxybenzoic, *p*-hydroxyphenylacetic, *p*-hydroxyphenylpropionic, and homovanillic acids.

### 4.3. Serum Samples

The concentration of the aromatic metabolites was determined in the serum samples from the critically ill patients and healthy volunteers; the total number of the serum samples was 196, the total number of subjects were 131. The serum samples were stored at −30 °C in the Federal Research and Clinical Center of Intensive Care Medicine and Rehabilitology (Moscow, Russia). The approval of the local Ethics Committee was obtained (N H 01/18, 12 July 2018). To study the samples from the patients (*n* = 79), admitted to the ICUs, serum residues taken for routine biochemical tests were used. All of these patients with severe traumatic brain injury [11], community-acquired pneumonia [13,14], or acute surgical diseases of abdominal organs [13,15] were previously, clinically described in the relevant studies. More than one serum sample in the dynamics was collected from 43 patients (2 samples from 28 patients, 3 samples from 9 patients, 4 samples from 5 patients, 5 samples from 1 patient). The normal values of the analytes were determined in the serum samples from healthy volunteers aged over 18 (*n* = 52) which were collected in Rogachev National Research Center (Moscow, Russia) [21]. The healthy volunteers were clinically examined before the blood donation; all of them met the necessary criteria. All of the serum samples (52 samples from healthy volunteers and 144 samples from critically ill patients) were collected into the tubes without preservatives, coagulation activators, or other reagents, kept at −40 °C, then defrosted at room temperature before the experiments.

### 4.4. Data Processing

The data processing was performed in the R environment (4.1.2) using RStudio. The statistical tests were carried out using the base R. The plots were generated using the ggplot2 package [28]; multivariate analysis was conducted using the mdatools package [29]; ROC analysis was performed using the pROC package [30]. The DUPLEX algorithm [31] was implemented by us in C and integrated into R, using the Rcpp package [32].

To predict the outcome of the critically ill patients, a classification model was built using SIMCA. SIMCA is a supervised classification method based on principal component analysis (PCA). In this work, we are solving a binary classification problem because our goal is to predict one out of two possible outcomes for a critically ill patient. In this case, each class can be modeled using SIMCA, however, only one class (containing both healthy volunteers and survivors) was modeled. The samples obtained from non-survivors were not considered during the model construction.

If several samples were obtained from the same patient, only one was used during the model construction. Outlier detection was the first step of data processing. It was based on both examining the concentration distributions and using multivariate approaches [22]. Similar results were obtained in both cases. If the concentration of at least one of the metabolites was atypically high, a sample was found to be an outlier. Initially, the samples taken on the day of admission were chosen for all of the critically ill patients. However, if any sample was found to be an outlier, it was replaced with another one taken a few days after admission. If such a sample was not available, the respective patient was not considered during model construction. As a result, seven samples taken on the day of admission were replaced and eight samples were excluded from consideration. If a sample taken on the day of admission was not found to be an outlier, other samples taken from the same patient were not tested to be outliers and were placed in the “Survivors (others)” group.

The total number of samples in the modeled class was 88. These samples were split in a ratio of 2:1 into two subsets, using the DUPLEX algorithm (healthy volunteers and survivors were treated independently) [31]. The first subset contained 59 samples. It was used to build and validate the classification model. A matrix containing relative areas of chromatographic peaks was used as an input. The autoscaling was performed before PCA decomposition. The number of principal components (*n*_(PCs)_ = 3) and the decision rule (for a selected significance level α = 0.01) were found, using the data-driven approach [23,33]. The validation was performed using the Procrustes cross-validation [34]. The second subset contained 29 samples and formed an independent test set. It was used only for the evaluation of the model efficiency.

All of the samples were placed in one of the following groups (without duplication):

“Calibration” group (*n* = 59) was the calibration set;

“Test” group (*n* = 29) was the test set;

“Survivors (outliers)” group (*n* = 28) contained all of the samples which were found to be outliers;

“Survivors (others)” group (*n* = 29) contained all of the remaining samples taken from survivors (i.e., all of the samples which were not included in the “Calibration”, “Test” or “Survivors (outliers)” groups);

“Non-survivors” group (*n* = 35) contained all of the samples taken from the non-survivors on the day of admission;

“Non-survivors (others)” group (*n* = 16) contained all of the samples taken from the non-survivors a few days after admission.

## 5. Conclusions

The use of metabolomics in discovering the alterations between patient groups, including critically ill patients, is a powerful tool, and a predictive model based on different metabolites can help in patient stratification. Despite the widespread use of generally accepted multi-parameter scales, such as the APACHE II clinical score, aromatic metabolites (one or a number of them) may be used in clinical practice to reliably predict the outcome of critically ill patients on the day of admission to the ICU. Our data showed that a univariate model based on *p*-HPhLA is able to classify correctly more survivors (less false-positive samples) than the multivariate models. An imbalance in the profile of aromatic metabolites in the blood of critically ill patients indicates serious metabolic disorders, which may have to become a new therapeutic target.

## Figures and Tables

**Figure 1 molecules-27-04784-f001:**
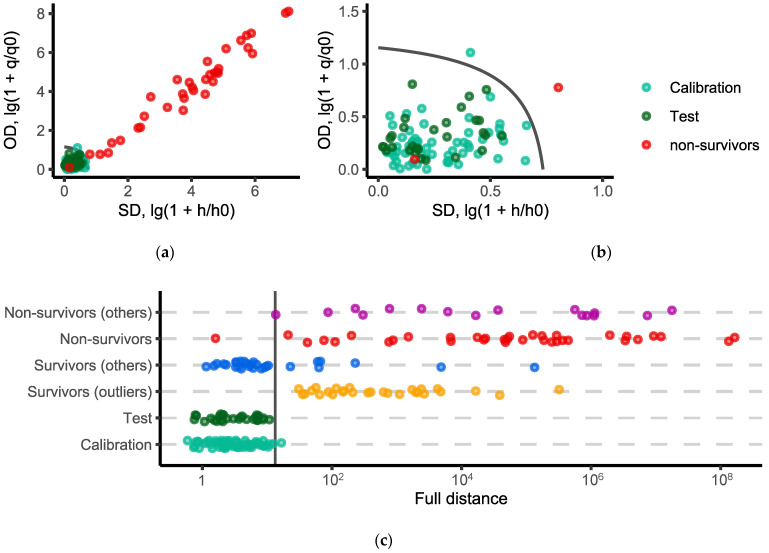
Distance plot (**a**) in original and (**b**) zoomed scale; and (**c**) one-dimensional visualization of the results obtained with SIMCA (points are scattered along the y-axis to improve perception in the case of overlapping). The black solid line is the decision rule (α = 0.01).

**Figure 2 molecules-27-04784-f002:**
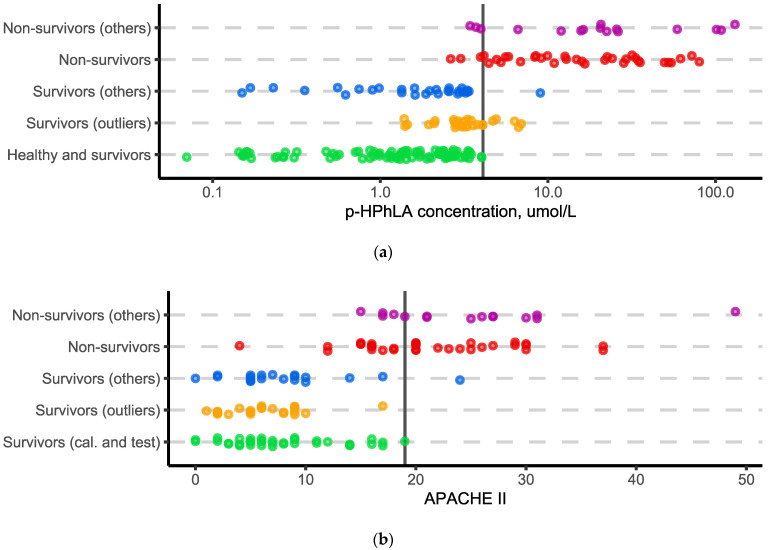
Visualization of the results obtained using the univariate models based on the concentration of the only *p*-HPhLA (**a**) and clinical APACHE II scale (**b**). The vertical line is the decision rule corresponding to the *p*-HPhLA concentration of 4.1 µmol/L or to the APACHE II score of 19 points. Points are scattered along the y-axis to improve perception in the case of overlapping.

**Figure 3 molecules-27-04784-f003:**
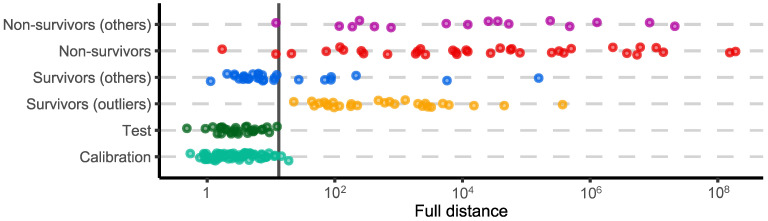
One-dimensional visualization of the results obtained with SIMCA. The model was built without *p*-HPhLA. The vertical line is the decision rule (α = 0.01). Points are scattered along the y-axis to improve perception in the case of overlapping.

**Figure 4 molecules-27-04784-f004:**
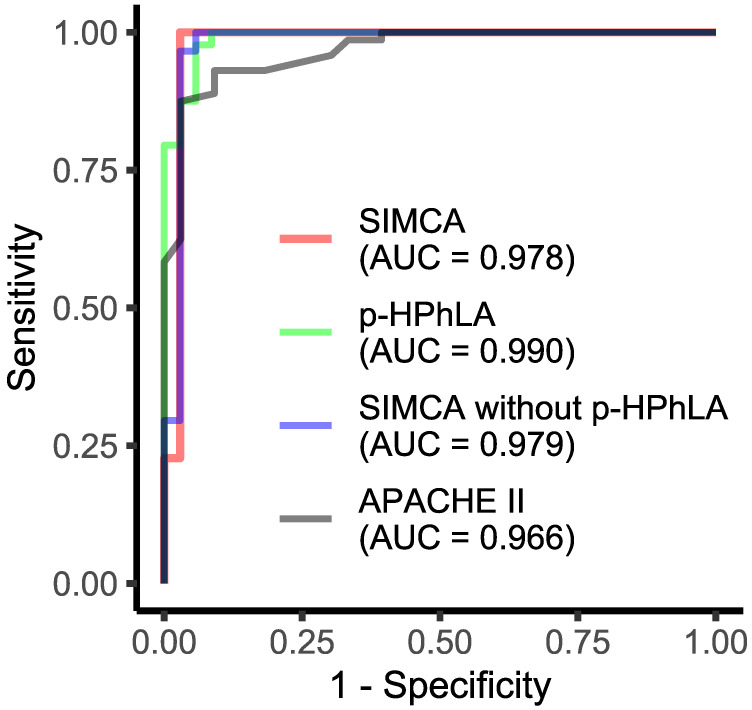
ROC curves obtained for univariate models based on the *p*-HPhLA concentration and clinical APACHE II scale, and multivariate models based on the profile of the 8 aromatic metabolites, including *p*-HPhLA, and on the profile of the 7 aromatic metabolites without *p*-HPhLA.

**Table 1 molecules-27-04784-t001:** Chemical structures of the aromatic acids, which are the metabolites of phenylalanine and tyrosine.

Amino Acid	Metabolites, Acids
Phenylalanine	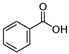	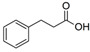	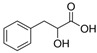		
Benzoic	Phenylpropionic	Phenyllactic		
Tyrosine	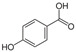	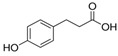	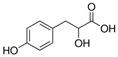	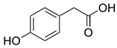	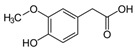
*p*-Hydroxy-benzoic	*p*-Hydroxyphenylpropionic	*p*-Hydroxyphenyllactic (*p*-HPhLA)	*p*-Hydroxyphenylacetic	Homovanillic

**Table 2 molecules-27-04784-t002:** Characteristics of the critically ill patients (*n* = 79), and the results of the clinical and biochemical analysis of their serum samples taken on the day of admission to the intensive care unit. Data are presented as median and interquartile range (25–75%), for vasopressors–% of the prescriptions. For each parameter *n* indicates the number of patients in whom this parameter has been measured. The results for the Mann–Whitney U-test for the comparison of the independent groups of survivors and non-survivors, which are significant at *p* < 0.05, are highlighted in bold. The Holm–Bonferroni method was used to adjust *p*-values for multiple comparisons.

Parameters	Normal Values	Patients		2-Tailed *p*
		Survivors (*n* = 44)	Non-Survivors (*n* = 35)	
APACHE II, points	0	7 (5–11) *n* = 43	20 (16–26) *n* = 33	**<0.0001**
SOFA, points	0	2 (1–5) *n* = 43	10 (8–13) *n* = 33	**<0.0001**
Glasgow Coma Scale, points	15	15 (13–15) *n* = 44	15 (13–15) *n* = 33	0.578
Oxygenation PaO_2_/FiO_2_, mmHg	≥400	305 (267–385) *n* = 31	282 (152–361) *n* = 31	0.141
Platelets, ×10^3^/μL	≥150	241 (175–313) *n* = 44	166 (89–208) *n* = 33	**0.0052**
Bilirubin, μmol/L	<20	12.5 (8.4–14.9) *n* = 42	19.0 (13.1–31.4) *n* = 28	**0.0082**
Creatinine, μmol/L	<110	80 (67–113) *n* = 43	191 (151–263) *n* = 31	**<0.0001**
Vasopressors, %	No	13.6% *n* = 44	91.4% *n* = 35	Not available

**Table 3 molecules-27-04784-t003:** Results of the GC-MS analysis of serum samples from healthy volunteers (*n* = 52) and critically ill patients (*n* = 79) taken on the day of admission to the intensive care unit. Data are presented as median and interquartile range (25–75%). The results for the Kruskal–Wallis test for the independent groups of healthy volunteers, survivors, and non-survivors, which are significant at *p* < 0.05, are demonstrated. The Holm–Bonferroni method was used to adjust *p*-values for multiple comparisons. For each parameter *n* (c > LOQ) indicates the number of samples in which this parameter has been quantitatively measured. To perform the statistical test missing values (i.e., below the limit of detection) were replaced by the half of the minimum of non-missing values.

Aromatic Acid, μmol/L	LOD/ LOQ Values *	Healthy Volunteers (*n* = 52)	Patients		2-Tailed *p*
			Survivors (*n* = 44)	Non-Survivors (*n* = 35)	
Benzoic	**/0.7	1.3 (0.8–1.8) *n* (c > LOD) = 52 *n* (c > LOQ) = 44	1.9 (1.2–2.7) *n* (c > LOD) = 44 *n* (c > LOQ) = 40	2.5 (1.5–5.1) *n* (c > LOD) = 35 *n* (c > LOQ) = 31	0.0002
Phenylpropionic	0.01/ 0.59	<LOQ (<LOD–0.73) *n* (c > LOD) = 37 *n* (c > LOQ) = 19	<LOD (<LOD–<LOQ) *n* (c > LOD) = 19 *n* (c > LOQ) = 2	<LOD (<LOD–<LOQ) *n* (c > LOD) = 16 *n* (c > LOQ) = 2	0.0001
Phenyllactic	0.4/ 0.5	0.6 (<LOD–0.8) *n* (c > LOD) = 34 *n* (c > LOQ) = 29	1.1 (<LOD–1.7) *n* (c > LOD) = 32 *n* (c > LOQ) = 32	4.0 (2.5–7.0) *n* (c > LOD) = 35 *n* (c > LOQ) = 35	<0.0001
*p*-Hydroxybenzoic	0.2/ 0.6	<LOD (<LOD–<LOD) *n* (c > LOD) = 0 *n* (c > LOQ) = 0	<LOD (<LOD–<LOD) *n* (c > LOD) = 4 *n* (c > LOQ) = 2	<LOQ (<LOD–1.9) *n* (c > LOD) = 20 *n* (c > LOQ) = 16	<0.0001
Homovanillic	0.1/ 0.5	<LOD (<LOD–<LOD) *n* (c > LOD) = 1 *n* (c > LOQ) = 0	<LOD (<LOD–<LOD) *n* (c > LOD) = 9 *n* (c > LOQ) = 6	1.7 (<LOQ–6.3) *n* (c > LOD) = 26 *n* (c > LOQ) = 21	<0.0001
*p*-Hydroxyphenylacetic	0.1/ 0.6	<LOQ (<LOD–<LOQ) *n* (c > LOD) = 28 *n* (c > LOQ) = 2	<LOQ (<LOD–1.9) *n* (c > LOD) = 32 *n* (c > LOQ) = 21	9.6 (2.8–18.5) *n* (c > LOD) *=* 33 *n* (c > LOQ) = 29	<0.0001
*p*-Hydroxyphenylpropionic	0.1/ 0.5	<LOD (<LOD–<LOD) *n* (c > LOD) = 0 *n* (c > LOQ) = 0	<LOD (<LOD–<LOD) *n* (c > LOD) = 2 *n* (c > LOQ) = 1	<LOD (<LOD–<LOQ) *n* (c > LOD) = 9 *n* (c > LOQ) = 5	0.0002
*p*-HPhLA	**/0.5	2.1 (1.6–2.6) *n* (c > LOD) = 52 *n* (c > LOQ) = 51	2.9 (2.2–3.4) *n* (c > LOD) = 43 *n* (c > LOQ) = 43	14.8 (6.5–32.6) *n* (c > LOD) = 35 *n* (c > LOQ) = 35	<0.0001

* LOD—limit of detection; LOQ—limit of quantitation, ** the LOD values in serum samples cannot be measured because benzoic acid and *p*-HPhLA are ubiquitous in human blood including healthy volunteers.

## Data Availability

Not applicable.

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
