# Peer review of "Multivariate Prognostic Model for Predicting the Outcome of Critically Ill Patients Using the Aromatic Metabolites Detected by Gas Chromatography-Mass Spectrometry"

_molecules, 2022, doi:10.3390/molecules27154784_

Round 1
Reviewer 1 Report
I have reviewed the manuscript entitled " Multivariate Prognostic Model for Predicting the Outcome 2 of Critically Ill Patients Using the Aromatic Metabolites 3 Detected by Gas Chromatography Mass-Spectrometry". This study is developed a method for rapid identification of characteristic aromatic metabolites in critically ill patients. The paper could be published in Molecules after the minor revised.
1. Please provide the structures of all Standards mentioned in 4.1 section. And there were too many abbreviations in the text which is hardly readable and understandable, authors could using compounds 1-8 to reprensent those names.
2. In the section of 2.2, from the table 2, the concentrations of p-HPhAA were vary significantly among the non-survivors, but authors just mentioned the HVA and p-HPhLA, please explain the judgment standard.
3. The creatinine and bilirubin level should better be detected in empty-stomach, but the authours just mentioned a sentence as “samples taken on the day of admission were chosen for all critically ill patients” In the Line 366, do the results listed in this study were correct?
4. In the section of discussion, do the authors try to summarized the theoretical data which could recognize as the potential markers, such as the concentration of aromatic metabolites, to distinguish the conditons of critically ill patients, patients and healthy people?
Author Response
Dear Review,
Thank you very much for your interest in our manuscript!
We have prepared responses to your comments and made the necessary corrections to the text of the manuscript:
- Please provide the structures of all Standards mentioned in 4.1 section. And there were too many abbreviations in the text which is hardly readable and understandable, authors could using compounds 1-8 to reprensent those names.
Aromatic acid abbreviations have been omitted from the text, with the exception of p-HPhLA, as it is one of the main metabolites that is repeatedly mentioned in the text. The structures of aromatic acids are added to the text in the form of Table 1, with division into tyrosine and phenylalanine metabolites.
- In the section of 2.2, from the table 2, the concentrations of p-HPhAA were vary significantly among the non-survivors, but authors just mentioned the HVA and p-HPhLA, please explain the judgment standard.
The p-HPHAA concentration varied significantly among all patients and this information has been added to the text (lines 121-127).
- The creatinine and bilirubin level should better be detected in empty-stomach, but the authours just mentioned a sentence as “samples taken on the day of admission were chosen for all critically ill patients” In the Line 366, do the results listed in this study were correct?
The reviewer is right - the determination of these biochemical parameters is usually determined on an empty stomach. However, in the case of the ICU, we could not influence the time of blood sampling, because these parameters should be determined in the blood as soon as possible to assess the severity of the patient. In addition, there are studies showing that the time of blood sampling does not greatly affect the determination of these biomarkers (the deviation of results is within 10%) . Thus, we assume that the time of blood sampling did not greatly affect our data, especially given how much the results differed between our groups.
- In the section of discussion, do the authors try to summarized the theoretical data which could recognize as the potential markers, such as the concentration of aromatic metabolites, to distinguish the conditons of critically ill patients, patients and healthy people?
Unfortunately, we are not sure that we understood the reviewer's question correctly.
If the reviewer's question was about whether we can divide all objects into 3 different groups: healthy people, survivors or non-survivors then here is the answer:
Reviewer 2 Report
The research article "Multivariate prognostic model for predicting the outcome of critically ill patients using the aromatic metabolites detected by gas chromatography mass-spectrometry" by Pautova et al. will adequately complement the special issue on mass spectrometry in health sciences.
The authors present their research on several aromatic metabolites as novel markers of bacterial complications in ICU patients.
In general, the written English is very good, and the article is written in clear and understandable way. Article structure is according to the journal instructions, and the references are adequate and reflect the research area presented in the article.
Author Response
Dear Review,
Thank you very much for your interest in our manuscript.
This manuscript is a resubmission of an earlier submission. The following is a list of the peer review reports and author responses from that submission.